# Hand Rehabilitation and Telemonitoring through Smart Toys

**DOI:** 10.3390/s19245517

**Published:** 2019-12-13

**Authors:** N. Alberto Borghese, Jacopo Essenziale, Renato Mainetti, Elena Mancon, Rossella Pagliaro, Giorgio Pajardi

**Affiliations:** 1Department of Computer Science, AIS-Lab Laboratory of Applied Intelligent Systems, Università degli Studi di Milano, 20133 Milan, Italy; jacopo.essenziale@unimi.it (J.E.); renato.mainetti@unimi.it (R.M.); 2Hand Surgery Unit, San Giuseppe MultiMedica Hospital, 20123 Milan, Italy; elena.mancon@gmail.com (E.M.); fisioterapiamano.sangiuseppe@multimedica.it (R.P.); gpajardi@centrostudimano.it (G.P.)

**Keywords:** smart objects, pressure sensing, wireless communication, 3D printing, emotional design, hand rehabilitation, exergames, hand surgery

## Abstract

We describe here a platform for autonomous hand rehabilitation and telemonitoring of young patients. A toy embedding the electronics required to sense fingers pressure in different grasping modalities is the core element of this platform. The system has been realized following the user-centered design methodology taking into account stakeholder needs from start: clinicians require reliable measurements and the ability to get a picture remotely on rehabilitation progression; children have asked to interact with a pleasant and comfortable object that is easy to use, safe, and rewarding. These requirements are not antithetic, and considering both since the design phase has allowed the realization of a platform reliable to clinicians and keen to be used by young children.

## 1. Introduction

The hand, both morphologically and functionally, represents one of the main elements that characterize humankind and serves as the main tool by which to interact with the environment in daily life. High finger mobility and opposable thumbs allow us to pinch, grab, manipulate, and interact with objects, enabling us to effectively explore the world, proving to be an essential element of the learning process throughout our life cycle, from childhood to old age [1].

During pregnancy, embryogenesis of the upper extremities takes place between the fourth and eighth week after fertilization, and the majority of congenital anomalies of the upper extremity occurs during this period. These occur nearly twice in every 1000 children born [2], and they manifest under several typologies with varying degrees of seriousness. In the most severe cases, when a few or no fingers are present (Oligodactyly), current clinical practice is based on surgically transferring one or more toes to the hand to enable it to grasp objects for interaction. However, to achieve this goal, a long rehabilitation period is required. Rehabilitation is also required, although for a shorter time, for those children who, after domestic or outdoor accidents, suffer from injuries to their hands (from fractures to finger amputation), requiring clinical intervention.

Classical rehabilitation is constituted of a set of exercises aimed to increase finger strength, coordination, speed, and accuracy of movement. Moreover, for transfer surgery, such exercises are aimed also to integrate implanted fingers into a new body scheme [3]. To engage children, exercises are usually carried out through toys (e.g., building blocks), that are manipulated by children according to the exercises goal and under the instructions and supervision of a professional therapist in one-to-one sessions. Often, this requires commuting to rehabilitation centers, inflicting significant organizational and economical stress on children’s families [4].

Information and C ommunication Technology (ICT) has the potential to change this, as it can enable patients to carry out part of the rehabilitation program at home. In recent years, the development of low-cost tracking devices, like Nintendo Wii Balance Board (https://www.nintendo.com/consumer/downloads/wiiBalanceBoard.pdf) and Microsoft Kinect (https://developer.microsoft.com/en-us/windows/kinect), that have shown valid accuracy [5], and the distribution of professional-level game engines, usually without costs for educational institutions (e.g., Unity 3D (https://unity.com/), Unreal (https://www.unrealengine.com/)), has been recognized to have a large potentiality in the rehabilitation field, especially for children for whom videogames are a natural entertainment tool. This has spurred the development of several rehabilitation platforms based on exergames, in which patients move an avatar inside the graphical interactive game environment through the tracking device in use to fulfil some game task. The movements elicited are functional to rehabilitation.

The most advanced platforms of this kind enable autonomous exercising and supervision is offered by clinicians through the data acquired by the platform itself during training [6,7]. This is a form of telemonitoring, which is defined as the use of information and computer technology to remotely monitor some aspect of the user’s clinical state at his/her home. It is relatively new in medicine but it has quickly developed particularly for certain pathologies. For instance, heart is monitored remotely transmitting the patient’s ECG to the hospital, where it is automatically analyzed on the fly; if a possible condition related to heart failure is detected an alarm is triggered to the patient and to clinicians, saving lives [8]. Given its success, telemonitoring is spreading fast on many other clinical domains like autonomous rehabilitation.

Several platforms, generically based on exergames and external tracking devices, both commercial [9,10] and research prototypes (e.g., [11,12,13,14]), have been developed to support autonomous posture [12,15,16,17] and arms [11] rehabilitation, but no platform of this kind, as far as we know, has been developed yet for hand rehabilitation especially in the pediatric age. This domain, in fact, provides a three-fold challenge: tracking the motion of the fingers, measuring grasping force, and providing a system that can engage children. Several platforms are described in literature (see Background section), but none of them can integrate all these aspects. Moreover, most of them do not provide a quick way to tune the exercise difficulty according to the child current motor ability, a fundamental feature to ensure an adequate challenge level.

We show here how a complete hand at-home rehabilitation system, based on exergames, with telemonitoring capability, can be realized using a tablet and an adequately designed pressure sensing device that is used as tracking device inside the exergames. To this aim, a modular flexible architecture that can sense fingers pressure has been designed. It wirelessly transmits pressure data to the host device, typically a smartphone or a tablet, on which the exergames are run. The incorporation of emotional aspects in the design of the sensor case has allowed for high acceptability among children. At the end of each game session, the exercise data are sent to the cloud becoming available to evaluate the progress, to schedule recall visits at the hospital, and to select a proper set of exergames and set their difficulty according to the actual child’s capabilities.

The paper is organized as follows: Section 2 reviews the most promising approaches to hand rehabilitation and telemonitoring. Section 3 describes in detail the platform that is discussed in Section 4. Section 5 draws the final conclusions.

## 2. Related Research

The first attempts to use technology to support hand rehabilitation come from the robotic field where instrumented gloves [18,19,20,21,22] and exoskeletons [23,24,25,26] have been proposed, also in combination with exergames. These systems have proven to be very useful when dealing with patients who do not have enough muscle power to move their hands on their own. However, they limit the freedom of patients’ movement and alter proprioception and natural behavior of the hand [27]. Moreover, they are bulky, not easy to dress, costly, and are therefore restricted primarily to clinical use with post-stroke patients. Lastly, their dimension makes it difficult to use them in pediatric hand rehabilitation.

For these reasons, sensing devices that have not to be attached to hands are preferred. A possible approach is based on using cameras to survey movement. Motion capture systems, based on using passive markers to robustly identify the finger joints on the camera images, were first proposed [28]. Although this approach is potentially accurate, it is limited to the laboratory domain, as the set-up time required to attach the markers to the hand makes this approach unfeasible at home. Of more interest are markerless approaches to motion capture. Recently, the Leap Motion^TM^ [29,30] and RGB-D cameras (e.g., Kinect^TM^ [31]) have been introduced to track the human body. However, such systems require that relatively simple movements, well visible to the camera, be performed. Moreover, their accuracy is not adequate for capturing free grasping movements [32,33]. Lastly, they require that the user lies inside the working volume of the device, while children tend to move a lot while playing. For these reasons, such devices are not a good match for hand rehabilitation, especially at a pediatric age.

Solutions that do not have such problems are based on touch screen devices in general and on tablets in particular. Tablets can detect the movement of one or more fingers on its surface with high accuracy [34,35,36,37,38,39]. Combined with exergames, they allowsguiding the users in exercises aimed to recover hand dexterity. Most of these solutions allow setting an adequate degree of difficulty and in some cases provide the clinicians with a log of children activity. However, a clear definition of an adequate set of rehabilitation exercises and their quantitative evaluation is still largely missing. Moreover, neither force exercises can be provided with these approaches nor grasping of real objects.

More recently, to enable force exercises, a few attempts to embed sensors inside objects have been described. These objects are inspired by the Myogrip [40], Takei [41], Jamar [42], or Vigorimeter [43] handheld dynamometers, that are used to measure the maximum grasping force exerted squeezing their handle. All of these devices are meant for clinical use. A few attempts to provide equivalent measurements at home have been proposed. The Grip-ball system is constituted of an inflated ball containing a pressure sensor and the associated electronics [44]. A similar approach has been adopted in the Domo-grip system [45], in the SqueezeOrb device [46], the latter developed for human-machine interfacing, and in the Lokee smart ball [47] or the Gripable device [48]. However, such devices allow only the palm power grasp and not all the other finer prehension modalities typical of the human grasping repertoire [49].

Moreover, all of these sensing devices maintain a clinical-like structure, in which the tracking object is clearly distinguishable and associated with measurement. This has been recognized to be an obstacle to adoption especially for children and the idea of instrumenting objects of everyday life has been recently pursued. In CogWatch project [50] objects of everyday life (e.g., coffee machines, mugs) have been instrumented with inertial and contact wireless sensors to gather data that can be used to assess the correct sequence of gestures (e.g., preparing a coffee) to early detect cognitive decline. Sen.se [51] has pushed this approach further providing motion sensors that can be attached to any daily life object. However, such approaches do not allow recovering detailed information on hand interaction. A more suitable approach to hand rehabilitation has been pursued in the Caretoy project [52], that has developed wireless cylindrical toys made of elastomer with embedded sensors. The toys are composed of two soft air-filled chambers connected to a pressure sensor; a rigid case inside the toy contains the control unit. We leverage this approach to realize a flexible pressure-sensing device that can be used to measure fingers pressure in different grasping modalities giving to it the aspect of a toy. This device, combined with a tablet device and with suitable exergames, allows supporting the large variety of exercises required for hand rehabilitation.

## 3. Materials and Methods

### 3.1. Platform Specifications

To provide a valid solution, the starting point is eliciting the functional specifications by all stakeholders [53], who, in our case, are:

**Clinicians (physiotherapists, neurodevelopmental disorders therapists and surgeons):** They require a tool that can guide the child through adequate exercises, which can be adapted to child current motor abilities. Moreover, they require remote monitoring of the rehabilitation progression from the hospital.

**Patients (children):** They have to use the platform for exercising autonomously. The platform should be easy to use and fun at the same time. We aim here to children between 2 and 7 years old for which a maximum force range of 5 kg has been defined.

**Caregivers**: These are, preferentially, family members. They need to minimize travel to the hospital with their child and maintain a tie with the clinicians at the hospital to tune exercise and optimize recall visits.

We have to co-design the platform with all of these stakeholders. To realize a system that can be useful and functional for the end-users, we have adopted two parallel processes throughout the development phase: the Design Thinking Process [54] and Human-Centered Design [55,56]: the first is based on searching for possible solutions to the problem, keeping the needs of the users at the center of the design process, while goal of Human-Centered Design is to maximize the usability and user experience of the system. Accordingly, we have involved all the users early in the design loop to take care effectively of several key interaction aspects: ergonomics, ease of use, and maintenance of the system, as well as effectiveness.

In the first phase, we have identified a core set of exercises required by observing live sessions and through focus groups with clinicians. These belong to two categories:

**Strength exercises**: Aim at strengthening fingers muscles to regain the power required to grasp objects. To achieve this, patients are required to grasp objects using different prehension modalities: pinch, palm, or side grasp [49]. Typically, therapists set a specific force, duration and intensity of the exercises and a repetition frequency.

**Mobility exercises**: Aim at regaining mobility, increasing sensitivity and range of fingers motion. To this aim, patients are required to tap, slide or pinch (e.g., like in the zoom-in/out functionality of many software applications) on a surface using one or multiple fingers. Here, therapists typically set the range of motion, its speed and accuracy [57].

We explicitly remark that the degree of difficulty of each exercise is regulated upon the current child’s motor ability.

The two sets of exercises call for different hardware requirements. The first set requires hardware capable of measuring finger strength when the user interacts with objects with different finger arrangements according to the different grasping modalities. The second set requires hardware capable of precisely track fingers position.

A single instrument cannot satisfy the two requirements in an unobtrusive and ecological way. If mobility exercises can be performed on the tablet screen as shown in [34,35,36,37,38,39], a sensing object that can be used for strength exercises was not available and has been developed as described here.

### 3.2. Platform Design and Development

#### 3.2.1. Force Sensing Sub-System

The core element of the platform is a specifically designed toy that embeds the dedicated hardware capable of sampling, digitizing and wirelessly transmitting in real-time pressure measurements to the mobile device (typically a tablet or a smartphone) on which the exergames are played (Figure 1). The toy realized here is based on Lego^TM^ that is a typical construction familiar to children of young age, and has the shape of a cottage (Figure 2a) that hides the sensing architecture.

The pressure sensor adopted should be robust enough to resist possible damage when handled by children. To this aim, we have adopted a load cell (13 mm × 13 mm × 50 mm) obtained disassembling a consumer kitchen electronic scale (Zheben, SF-400) with a pressure range of 0–7 kg. The load cell has been encapsulated inside a Lego^TM^ structure and the sensitive part of the cell has been covered by a flat Lego^TM^ red tile (Figure 2c) to indicate clearly where the child has to press. The encapsulated load cell has then been positioned inside the Lego cottage in an area where the children can comfortably interact with.

The processing electronics is housed inside the cottage and connected to the load cell. The force signal is amplified by a differential operational amplifier with an adjustable gain (TI INA125P); a particular configuration has been adopted to maximize stability as shown in Appendix B. The amplifier is connected to the micro-controller of an Arduino board that samples the signal, filters it, and sends it wirelessly through a Bluetooth chip HC-06 transceiver [58] at a frequency of 250 Hertz. Bluetooth 2.0 + EDR (Enhanced Data Rate) has been adopted to limit current sink and therefore power consumption. To maximize attractiveness, children can personalize the cottage with Lego^TM^ characters or other constructions, provided that these do not interfere with the sensing area.

In the configuration shown in Figure 2a,b, the child can activate the sensor by pressing with one finger. To accommodate pinch grasp, a rotating mechanism, made of a Lego^TM^ shaft inserted into two parallel bricks, has been designed and realized (Figure 2c). In this way, the Lego^TM^ case of the load cell is free to rotate and emerges from its housing; the child can thus squeeze the case with two opposite fingers as required by pinching. Different pinch amplitudes can be accommodated by simply making the load cell case thicker by adding flat bricks on the top of it (Figure 2d). Finally, for power grasp (palm grasp), the cell case can be detached from the cottage and used as a free-tracker (Figure 2e). In this way, the entire repertoire of basic prehensions of the human hand is covered.

Two ranges of force have been defined: from 0 to 3 kg and from 0 to 5 kg, according to age and rehabilitation state. The prototype is built on a 30 cm × 30 cm Lego^TM^ base and the weight of the construction is about 0.3 kg, which makes it easily portable. The accuracy and reliability have been fully tested as reported in Appendix C. Linearity and repeatability are in the order of 0.1%, which is 3 g for the 3 kg range and 5 g for the 5 kg range.

Such a high resolution is not required when the child exercises at home where the capability of applying a force above a given threshold is sufficient for training purposes. Under this hypothesis, the tracker can be simplified and a simpler sensing mechanism, based on the same principle on which a clothespin works, has been devised.

The sensing object is constituted of a pet-toy realized through 3D printing (FABTotum Personal Fabricator printer). This has been designed to be attractive to children: a crocodile made of two parts: the upper and lower jaw pivoting around the end of the mouth (Figure 3). The child has to press the tail of the crocodile to make it open its mouth; such movement is resisted by a set of rubber bands applied to the mouth whose strength depends on their number and regulates the force that has to be exerted by the child. A smart button, constituted of a digital contact, a microcontroller, and a radio transmitter, is inserted through a frontal slot inside a lodge created in the bottom jaw of the crocodile, to make battery substitution easy. The button is used to sense when the user has exerted enough pressure to open the mouth. The smart button chosen here is a Camkix Bluetooth Remote Shutter; typically used as a remote shutter of digital cameras, it has the robustness required for use by children. Such a device transmits a digital pulse whenever the contact is open and is compatible with the blue-tooth HID (Human Interface Device) profile; it can be easily interfaced with smartphones and tablets operating system as it is recognized automatically, like regular keyboards or mice, and can therefore be used easily as an input device.

#### 3.2.2. Motion Sensing Sub-System

For the mobility exercises, we leverage the multi-touch display of mobile terminals like smartphones or tablets. Such devices accurately detect the position of one or more fingers at 30 Hz and can therefore easily be used to measure fingers tapping, sliding, or pinching over the device surface (see videos in the Appendix A). This tablet or smartphone will be the same that works as a host for the pressure-sensing device.

#### 3.2.3. Platform Architecture

The sensing devices and the host constitute the platform that is given to the families to support autonomous rehabilitation of their child. It can be regarded as the client component of the typical client-server architecture used in the telerehabilitation domain [6]: the client runs the exergames, acquires the pressure data from the sensing devices, logs them and attaches a time stamp. At the end of the exercise, it transmits these data to a server, that, in turn, processes the data to compute the results and show them to the clinicians. Moreover, the server sends to the client the list of the exergames chosen by the clinicians, along with their difficulty degree (Figure 4).

The server is composed itself of two main components: the data storage and a graphical application that allows clinicians to analyze rehabilitation progression and to configure the rehabilitation sessions. Such a graphical application is implemented as a web-application to allow clinicians to access the data from anywhere, even outside the hospital.

When the application is started on the mobile terminal, the user sees the exercises grouped into a suite: the user can choose among the different exergames assigned by the therapist. The exercises will be already configured at the right difficulty and to be played either with the tablet or with the pressure sensor as a tracker (Figure 5).

### 3.3. Therapeutic Exergames

A set of exergames were developed, following a methodology specifically designed for therapeutic exergames [53], that consists of three main steps:

**Identification of the exercises**, all their requirements, constraints and all the parameters that determine their challenge level;

**Transformation of the exercises into Virtual Exercises** to discuss with clinicians game mechanics and actions;

**Introducing all the contour elements** that characterize a video-game (characters, graphical elements, color, music, sounds and rewards) as a layer on top of the exercises themselves to maximize entertainment.

The exergames were designed to implement different exercises such that the same game can be played to increase either force or mobility, thus creating a multi to multi mapping between exercises and exergames. We briefly describe here some of the developed exergames. As required, they cover the range of 2–7 years old.

#### 3.3.1. Breaking Eggs

A closed egg is placed in the middle of the screen. The player has to scratch the egg surface to break it. As a reward, a pet animal, different from trial to trial, exits from the egg and appears on the screen; the child can caress the pet, receiving an additional reward from the game: cute little stars or hearths, and pleasant sounds (Figure 6a). This game supports both mobility and strength exercises. The child can break the egg by tapping, sliding, or pinching on the display surface (mobility exercises), or by pressing, pinching, or grasping the sensor (strength exercises). The simple rewarding mechanism of the game has been designed for children 2–3 years old.

#### 3.3.2. Jigsaw

A nine-piece jigsaw is proposed to the child, who can solve it by dragging each piece in the right place with his/her hand. This game supports all the mobility exercises and can be played only through the multi-touch screen (Figure 6b). Different jigsaws are provided, and additional ones can be obtained from images provided by the child himself. It has been designed for children 3–7 years old.

#### 3.3.3. Mouse & Cheese Maze

A mouse is trapped inside a maze and the child has to guide it with his/her hand all the way to a target where it finds a piece of cheese. The maze shape is created procedurally and will be different from trial to trial to increase variability. This game supports all the mobility exercises and can be played only through the multi-touch screen (Figure 6c). It has been designed for children 3–7 years old.

#### 3.3.4. Hot Air Balloon (and Similar Themes)

This is a runner game. The player is in control of a hot air balloon that flies in the sky. The goal is to catch targets (e.g., coins) and avoid distractors (e.g., birds) by moving the balloon up and down by moving his/her fingers according to the specified exercise (Figure 6d). This game supports all the exercises using either the multi-touch screen or the pressure sensing device. Some variants of this exergame are available. In a marine theme, the graphical environment is the sea, the controlled element is a submarine, the targets are coins, and the distractors are fishes. In an outer space theme, the graphical elements represent the stars, the mobile element is an astronaut, and the distractors are aliens. The different themes allow showing to the child games that appear very different one from the other, while they share the same mechanics and guide the child through the same exercises. These games have been designed for children of 4–7 years old.

## 4. Discussion

To guarantee a good, fast, and complete recovery, rehabilitation after surgery should be continuous and intensive [59], and the system described here goes in this direction: it is meant to be used for a prolonged time and at home, autonomously. For this reason, a careful design phase is required. We have started from the clinical needs and developed the sensing part first. The touch screen of mobile devices has been identified as a suitable means to track the movements required to increase finger mobility. Pressure detection instead has required the design and development of a novel flexible pressure-sensing device.

We have started designing the first prototype to illustrate to clinicians the sensing mechanism [53]. This was constituted of the bare load cell attached to a rigid metal frame (Figure 7a) and to a simple clothespin with a button that detected when the clothespin was closed (Figure 7b). The load cell was considered adequate, as it showed enough robustness for children use; the clothespin mechanism was considered suited to elicit an adequate force from the children, but the grasping region on top was considered somehow slippery and a better shaping was required for grasping.

The second step was to provide a case that can be attractive to children. For this reason, we have embedded the load cell into a Lego^TM^ cottage structure (Figure 7c). This has allowed on one side to hide the sensor inside the cottage, and on the other side to offer to the child an object with whom he/she can be familiar. Following a similar line of thought, the clothespin was transformed into a 3D pet animal. A crocodile was chosen as it has two desirable characteristics that are a good match for the sensor requirements: it has a large head, inside which the trigger sensor can be hidden, and it has a long tail that can be used to induce the grasping movements required by clinicians in an ecological way (Figure 7d). Additionally, the tail has been designed with scales that facilitate a firm grasp. Moreover, it has allowed us to greatly simplify the case, reducing cost and size.

The load-cell based sensing device has first been tested for accuracy, repeatability, and linearity (Appendix C) to assess measurement validity following a procedure similar to that proposed in [45]. The resolution and accuracy are on average an order of magnitude larger than that of dynamometers typically used in the clinics [40,41,42], that have an accuracy of 50 g, with a sensitivity of 10 g [40]. This is due to the fact that such dynamometers have been developed to assess grip force mainly in adults and elders, and have a range that is almost twenty times larger as it reaches 90 kg. The device itself is also of large dimension. Focus on the pediatric hand has allowed miniaturizing the device on one side and increasing its accuracy and resolution on the other, to values that are larger than the target set by clinicians.

A single non-rechargeable button cell (CR2032 3 V) is used in the crocodile sensor. This is dimensioned to operate the device for one month, half an hour a day before the cell discharges and has to be replaced. The opening in the lower jaw (Figure 7d) makes this operation easy. The Lego^TM^ cottage accommodates a larger source of energy inside the case. We use here a pair of rechargeable stylus batteries (1.5 V AA) that are recharged typically at the end of the week. Indeed, recharging modalities that do not require extracting the batteries from their housing (USB, induction) would be preferable, but they require additional room for the charging components and will be part of a subsequent refinement stage.

The two sensing systems were then tested with stakeholders. Clinicians have validated the adequacy of the interaction modalities offered and of the measurements collected. Children, on their side, generally liked the objects; however, their feeling and that of their parents was that the sensing systems were somehow “cold” objects, not really interesting for children per sè.

Eliciting some emotional reaction is fundamental to increase the attractiveness and therefore intrinsic motivation to use the system. To this aim, we have added some emotional content to the objects [60]. We have added Lego^TM^ characters and additional aesthetic Lego^TM^ structures on top of the cottage (Figure 2). Moreover, we have also granted to children the possibility to further personalize the cottage. The crocodile’s surface was pained with a-toxic acrylic painting to provide a colorful and playful final appearance (Figure 3). The degree of pleasantness was largely increased in this last step and children were keen to use the crocodile repeatedly [61].

Although the overall system is quite simple, it contains one of the main requisites of telerehabilitation and telemonitoring systems in general: they can be adjusted to the patient’s current capabilities [6,7]. The combination of the exergames with the sensing devices enables a difficulty progression in the exercises along the three dimensions typically required by clinicians: range, speed, and accuracy. In all exergames, the game pace sets the speed, which is the frequency of spawning targets and distractors. The therapist can then program the range of motion required to consider a movement valid. For instance, for pinch movement, the child has to open the fingers by more than 40 mm, or the pinch force has to be larger than 1 kg. Only valid movements allow moving the avatar inside the game. Finally, the degree of accuracy can be increased by reducing the dimension of the targets to be hit by the game avatar. Similar reasoning applies to the other grasping modalities. Such flexibility in the platform is allowed by a combination of an adequate range and resolution of the sensors with the definition of specific and intuitive game parameters that regulate the exercise difficulty, according to the good practice of serious games design [12].

A single suite of exer-games is provided to guide both exercises on mobility and strength. This allows speeding up game learning and increases the attachment to the graphical characters that are used again and again. The exergames are designed to automatically collect data from the input devices (position or force), as well as game interaction outcomes (targets hit/miss, distractors hit/miss) along with their timestamp. Force data depend on the sensor used. They are values expressed in grams for the cottage sensor device and they are binary values (trigger values) for the crocodile toy; the latter notifies the game that the user applied enough force on the sensor. Position data are obtained by converting fingers position expressed in pixels into millimeters, considering the device-specific screen resolution (PPI: pixels per inch). These data are used by the game itself to evaluate user movement, interact with game elements, and provide feedback in real-time. When a game ends, the data acquired are sent to a remote server in the cloud for storage and analysis by the therapists who can access them through a web application. This modality allows them to review the rehabilitation progression of their patients in the most suitable period of the day. Moreover, this allows creating a loop that starts from the therapist, who assigns the exercises to the patients with a predefined level of difficulty, goes through the exercises that are carried out at home, and goes back to the clinicians with the results that can be used to tune the therapy. Such continuous telemonitoring capability is at the basis of any autonomous rehabilitation platform.

To make therapist evaluation most effective, views of the data with a different level of detail are provided. To get a quick and clear picture of patient progression, data are aggregated on a per-exercise basis according to the methodology described in [62] and average values on speed, accuracy, and range of motion are plotted over time. For instance, a typical plot of pinch progression is shown in Figure 8.

The clinicians can also analyze in-depth a single exercise of a single rehabilitation session through a detailed report like that shown in Figure 9.

This in-depth analysis of the time course of movement and force development along with game interaction outcome allows having a clear picture of how the exergame has been played. All these data, collected over the cohort of patients served by the same hospital, can be processed through adequate machine learning techniques to cluster patients, identify suitable models of rehabilitation progression for each cluster, and optimize the scheduling of hospital recall visits and the therapy program [6].

Game results, such as score, number of target, or distractors hit and missed are also displayed in this report; these values are not used by therapists to assess exercise performance quality, but to establish whether the game is at the right difficulty level. If the game becomes too simple or too difficult, the patient tends to be distressed by the game and quit [12,53]. This information is used to set a proper challenge level for the user with the goal of keeping him/her interested and engaged in the game itself (flow state).

Indeed, the score achieved in a game is dependent not only on the use of hand/fingers, but also on the gamer’s skill that theoretically can improve by continuing with playing. However, this possibility is very remote here as the ability in the game is strictly dependent on the force or motion skills acquired. Therefore, this information is used to set proper level of difficulty of the exer-game, by increasing range, force, or accuracy required to animate the avatar.

Overall, the platform presented here is a general approach in which sensors disappear inside objects and make the objects smart as they can measure some relevant aspect of everyday life in a real setting. Such smart objects are the basis of what can be classified a telehealth rehabilitation service in which the toys developed are integrated inside a cloud-based system that provides the analysis of rehabilitation data and relevant information to clinicians [62].

The platform can acquire a huge amount of data from a large population of patients. This enables machine learning algorithms to browse the data to extract meaningful information at the population level. One of the most promising directions is the identification of homogenous clusters of patients for whom a tailored rehabilitation program can be defined and the frequency of recall visits can be optimized. The same data can allow the comparison of different rehabilitation programs to identify strong and weak elements.

The large population could also be the basis for creating a virtual community of patients and their parents. This can, on one side, support gamification to boost child motivation in performing the exercises (e.g., virtual medals, points, leader boards, ranking, and so forth, cf. [63]), while on the other side it connects all the stakeholders and provides them a place where to exchange information, data, suggestions, and where they can find continuous support from peers.

Additional work can be spent in trying to identifying automatically when the child has made the right movement. In fact, children tend to use either their intact fingers or their intact hand to perform the movements required by the exergames; presently, their parents correct them to perform the right movement. This could be achieved, for instance, by analyzing the pressure or movement time course with adequate classification algorithms. Finally, although the hardware is relatively simple and cheap (at present less than 100 Euros), its cost should be further reduced to enable massive diffusion.

## 5. Conclusions

We show here how combining electronics with functional and emotional design, a sensor, which can be used for clinical purposes, can be realized such that is appears as an attractive toy to its young users. This allows prolonged natural use that is the goal of such smart objects and opens the way to increased contamination between mechanics, computer science, electronics, and design to produce objects of everyday use that can have transparent sensing capabilities. Such an approach also allows the combining of telerehabilitation and telemonitoring into a single platform, thus making the system particularly suitable for autonomous use at home.

## Figures and Tables

**Figure 1 sensors-19-05517-f001:**
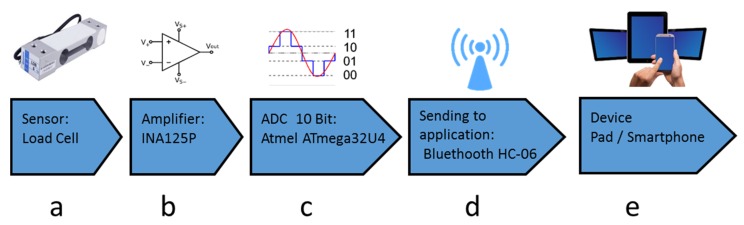
The modular sensor architecture, embedded inside a toy, is represented here. (**a**) Force is measured at the prescribed contact point of the object through a load cell. (**b**) The signal is amplified, (**c**) sampled and filtered by a micro-controller and (**d**) transmitted to a remote host, where (**e**) it is used to animate an avatar inside a graphical game environment (cf. Figure 2a).

**Figure 2 sensors-19-05517-f002:**
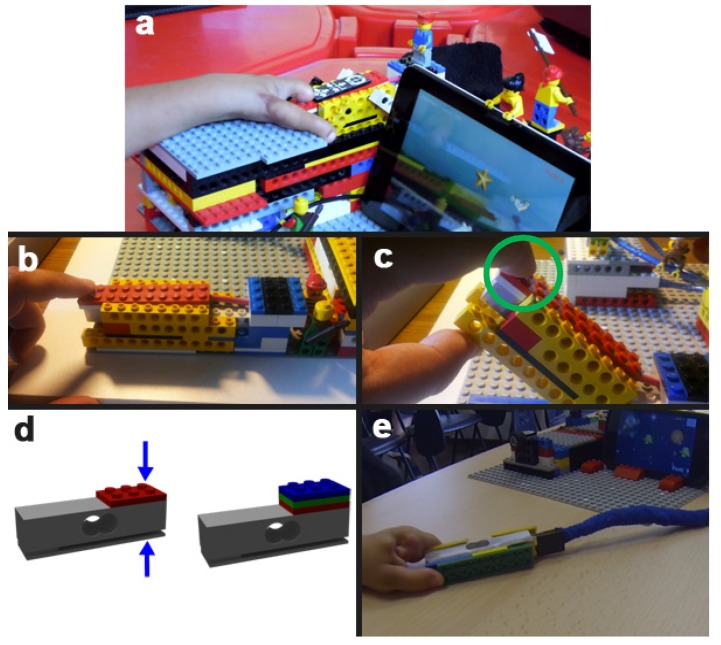
(**a**) The toy developed to measure force is shown here. A personalized Lego^TM^ construction hosts the sensor and the associated hardware. Notice that all the hardware components are not visible. The output of the sensing device is wirelessly connected to the tablet on which the game is run. (**b**) A zoom of the housing of the load cell, covered with Lego^TM^ bricks. (**c**) A Lego^TM^ shaft allows raising the cell such that pinch movements are also allowed. The flat tile covering the pressure point is highlighted in green. (**d**) The same structure can be used to accommodate different fingers apertures (depending on age, rehabilitation stage etc.). (**e**) The sensor and its Lego^TM^ case can be detached from the housing and used as a free-sensor for more complex grasping movements.

**Figure 3 sensors-19-05517-f003:**
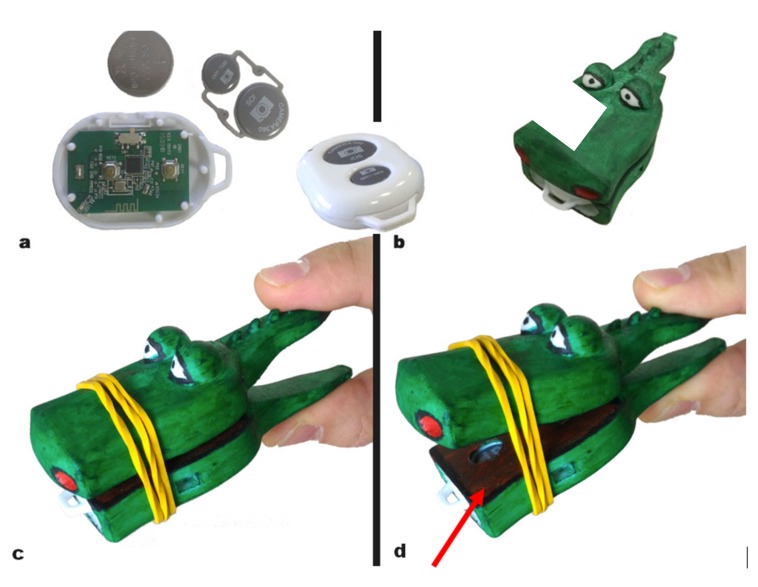
A remote shutter (**a**) is inserted inside the lower jaw of the crocodile (**b**) Force has to be exerted on the tail to open the mouth (**c**) When the mouth is closed a tooth in the upper jaw presses against the button trigger (grey area, red arrow, **d**), while when the mouth is open the tooth is raised and the button is releasedThe amount of force is regulated through a rubber band. The crocodile’s size is: 100 mm × 45 mm × 35 mm and weighs about 50 g.

**Figure 4 sensors-19-05517-f004:**
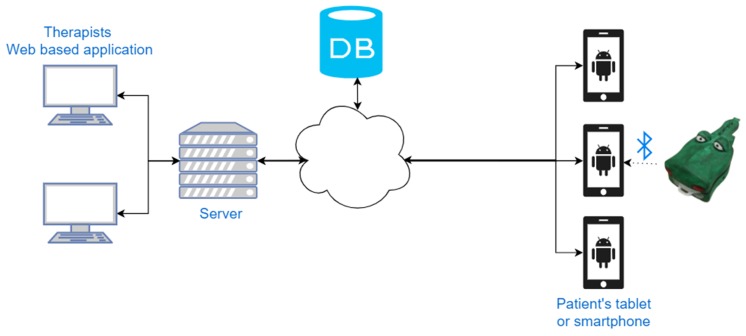
The whole system architecture is shown. It consists of the following components: a mobile terminal as a host on which the exergames are run. The host is connected wireless with the smart toy used as tracker and transmits to the server the data acquired during the interaction. Such data can then be accessed through a web application form any browser by the therapist.

**Figure 5 sensors-19-05517-f005:**
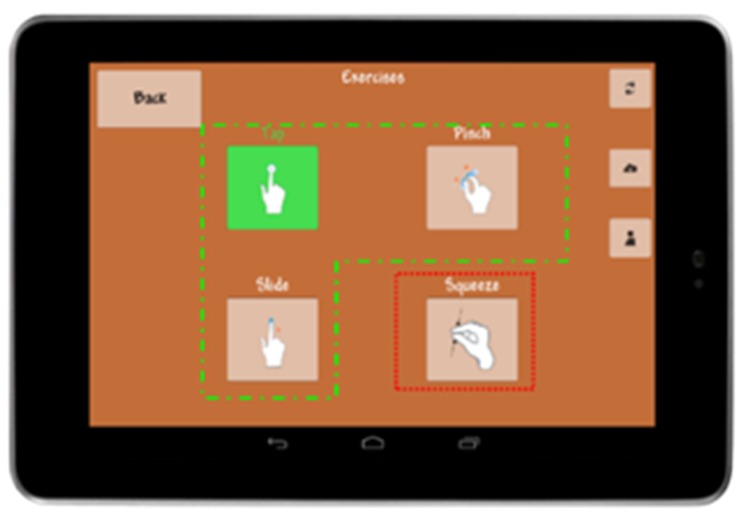
The exergames prescribed by the clinician are shown. They have already been configured at the correct difficulty level.

**Figure 6 sensors-19-05517-f006:**
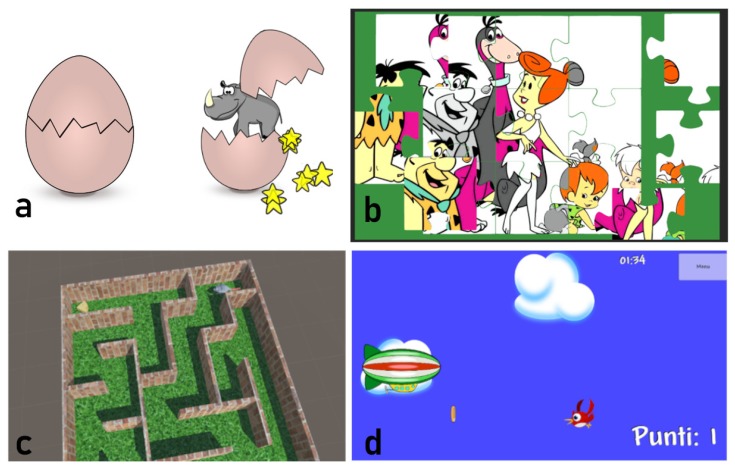
Different exergames developed for hand rehabilitation: (**a**) break-egg, (**b**) puzzle, (**c**) maze (**d**) hot air balloon. The mapping between games and exercises is multi to multi. For example, the pinch exercise can be guided by brake-egg games or hot air balloon; hot air balloon, on its side, can support all mobility and strength exercises.

**Figure 7 sensors-19-05517-f007:**
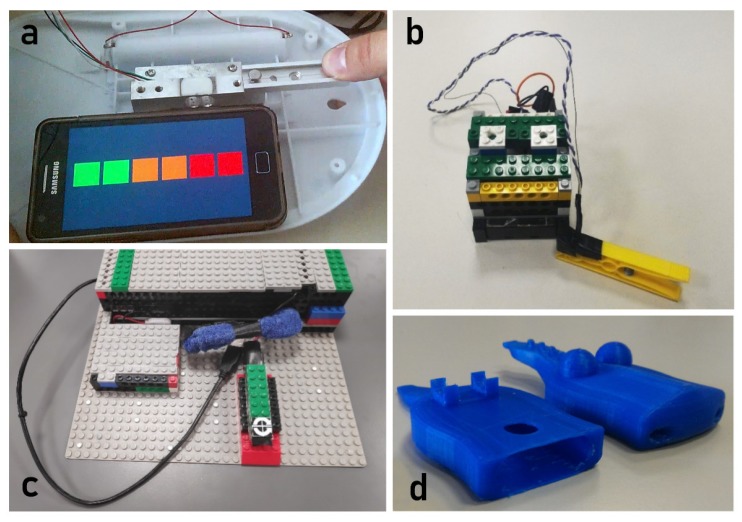
(**a**) The first prototype of the load cell-based sensing device. No case is provided to the bare sensor. (**b**) The first prototype of the clothespin based sensing device with an embedded sensor that detects when the clothespin is closed. Such prototypes are enhanced in the lower row. (**c**) The second prototype of the load-cell based toy capable of detecting the force. All the electronics is enclosed within the two Lego^TM^ cases, which are kept very basic. (**d**) The clothespin has been turned into a crocodile and the sensor and the electronics are accommodated inside the crocodile lower jaw and the Lego^TM^ case eliminated. No painting of the crocodile is provided.

**Figure 8 sensors-19-05517-f008:**
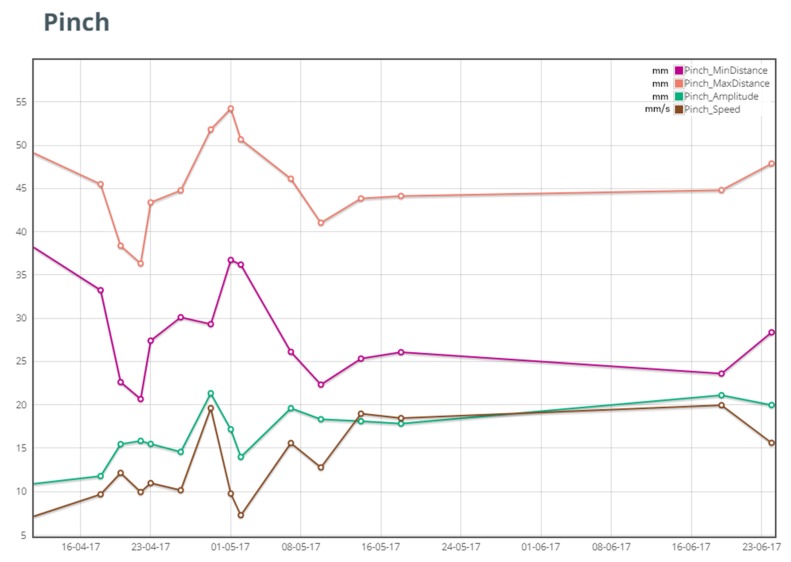
Pinch report on all pinching exercises performed in a 10 weeks period. The amplitude of pinching increases largely in the first month of training and then increases even more as shown by the green line. The same is true for pinch speed. Notice that a clearer picture can be obtained from the trend over time of the performance. Dots represent rehabilitation sessions carried out with the exergames.

**Figure 9 sensors-19-05517-f009:**
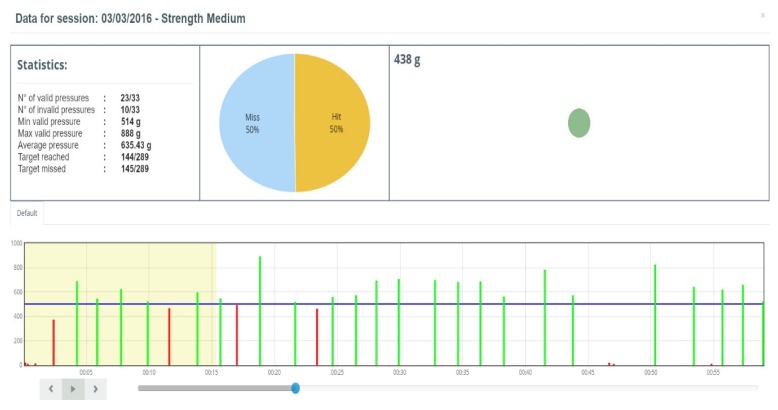
Detailed results of one pinching exercise carried out with the Lego^TM^ cottage. Here the force threshold to have a valid pinch was set to 0.5 kg (blue horizontal line). Numerical results are reported in the left panel. About two-thirds of the pinch movements were over-threshold. Each pinch is shown on the bottom. The therapist could review the rehabilitation session by clicking on the play button (bottom-left corner). During the animation, the diameter of the green dot on the upper right panel represents the force exerted during the current time instant. The yellow colored box in the bottom chart represents the elapsed frames from the beginning of the exercise to the current time instant.

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
