# Peer review of "Hand Rehabilitation and Telemonitoring through Smart Toys"

_sensors, 2019, doi:10.3390/s19245517_

Round 1

Reviewer 1 Report

The paper presents an embedded system to be used for the hand rehabilitation of young children. The system is composed of sensors to perform hand rehabilitation exercises in the form of video games played on the tablet and a wireless transmission system to transmit the collected data to a remote host for further processing.

My major concern with the paper is that, while the designed system is presented in details, almost no information is given on how data are processed. Some tablet games can be played using the load cell based sensors while others are played by touching the tablet screen with the fingers. What type of data are transferred to the remote host for analysis (data from the sensors, i.e. analog voltages, over time or only the results of game, i.e. win/lose, score, time to achive a goal etc.) and how these data are processed (machine learning algorithm etc.)? Moreover, the score achieved on a game is dependent not only on the use of hand/fingers but also on the gamers skill that is supposed to improve by continue playing. How the system can compensate this to estimate the improvement in hand/fingers use from the game data. Another major issue is the lack of comparison between the proposed technique and reference techniques for hand/fingers rehabilitation using a trainer on-site. Have the authors made experiments, for example using the proposed system on a group of young children and standard techniques for another group to compare the performances (hand/finger rehabilitation over time etc.) between the two methods? Other minor issues are: line 89, “)” is present but “(“ is not present; line 130, “between 2 and 7 years old” but “1-7 years old” is presenta t line 207; line 211, “eff” instead of “egg”; figure 5 is too small and almost not readable; figure 6 caption, “is clothespin is”; figure 7 is too small and almost not readable; line 396, there is a bad reference to “figure 2”, the reference is to the figure of page 13 (INA125 amplifier) that is not correctly referenced as figure 7 (fig 7 is already presenta t page 10) and should be referenced as fig. 8; figure 8 at page 14 should be refernced as figure 9; line 443, there is a not correct reference to figure 4.

Author Response

Resubmission of the paper: "Hand rehabilitation and tele monitoring through smart toys", by N. Alberto Borghese, Jacopo Essenziale, Renato Mainetti, Elena Mancon, Rossella Pagliaro, Giorgio Pajardi, for the Special Issue on “Sensor Technologies for Caring People with Disabilities for Sensors journal.

We have realized that in the first draft of the paper some information has been condensed too much. We have therefore revised deeply the organization of the paper, as suggested by Reviewer N.2, as follows:

-        Introduction has been split into introduction and background. Introduction contains the motivation and the problem addressed (rehabilitation and telemonitoring of paediatric hand). The last paragraph summarizes the content of the different Sections. Background. We revise the literature on the topic subdividing it into 4 categories: robotics, camera trackers, tablet and smart objects. The final paragraph clearly marks what is still missing in these approaches.

-        Materials and Methods is now organized according to a clear path: functional specifications definition, implementation of the platform and on a flexible sensing device in the for of a toy, the general architecture and the exergames defined. Section on specifications has been streamlined, with a clear definition of the methodology and of the exercises identified.

-        Discussion. A comparison of the platform with exisiting methods both in terms of accuracy and usability.

-        Conclusion that stresses the role of smart objects for telemonitoring and telerehabilitation.

-        Appendix has been split into Appendix A (Amplification design) and Appendix B (Characterization of force measurements). These contains details that the reader might be interested in. However, now the results of the Appendixes are fully reported and discussed in the main text. 

As reorganization and rewriting has been very heavy we could not use track change Word functionality.

Moreover, the text was proof-read by a collegue of us who has been living in the United States for two years. In this process, all small errors should have been corrected. If this were not sufficient and the paper were accepted, we will have the paper revised by a professional English reviewer.

We report here a point-to-point reply to reviewer N.1 concerns:

Reply to comments of Reviewer N. 1

Q. My major concern with the paper is that, while the designed system is presented in details, almost no information is given on how data are processed.

A. The reviewer correctly stresses that any tele-health platform is built around the clinical needs. We are well aware of this and have published a paper well quoted on this (ref. [53] of the paper). We now report on data processing in Section 3.1 Functional requirements, where the required exercises and their evaluation by clinicians is provided. We the report two instances of the data returned to clinicians in Figures 8 (where long time results are shown for one of the prehension modalities) and in Figure 9 (where a detailed picture of one exercise is provided). The architecture that collects the data, processes them and produces the plots is described in Section 3.2.4.

Q. Some tablet games can be played using the load cell based sensors while others are played by touching the tablet screen with the fingers. What type of data are transferred to the remote host for analysis (data from the sensors, i.e. analog voltages, over time

We have now better specified this aspect. Different digital values area acquired with the Lego and the crocodile toy. Both values receive a time stamp from the host that acquire them. We specify this in the following paragraph of the text:

- at the beginning of Section 3.2.2: “The core element of the platform is a specifically designed toy that embeds the dedicated hardware capable of sampling, digitizing and wirelessly transmitting in real-time pressure measurements to the mobile device (typically a tablet or a smartphone) on which the exergames are played (Figure 1).”

- At line 263 (just before Figure 3). “The button is used to sense when the user has exerted enough pressure to open the mouth. The smart button chosen here is a CAMKIX Bluetooth Remote Shutter typically used as a remote shutter of digital cameras and has the robustness required by children use. Such device transmits a digital pulse whenever the contact is open and it is compatible with the blue-tooth HID (Human Interface Device) profile; it can be easily interfaced with smartphones and tablets operating system as it is recognized automatically, like regular keyboards or mouse, and it can therefore be used easily as an input device.”

- At the beginning of Section 3.2.4: “the client runs the exergames, it acquires the pressure data from the sensing deveces and logs them and attach a time stamp.”

Q. or only the results of game, i.e. win/lose, score, time to achieve a goal etc.)

It is now specified in the paragraph at line 478 that :“Game results, such as score, number of target or distractors hit and missed, are also displayed in this report; these values are not used by therapists to assess exercise performance quality, but to establish whether the game is at the right difficulty level. If the game becomes too simple or too difficult, the patient tends to be distressed by the game and quit [12,53]. This information is used to set a proper challenge level for the user with the goal of keeping him/her interested and engaged in the game itself (flow state).”

and how these data are processed (machine learning algorithm etc.)?

In the simplest version we report results averaged over the same exercise according to the methodology introduced in Reference [62] that aims to providing quantitatinve over a set of different exercises that target the same rehabilitation goal. This is explained now before introducing Figure 8. Indeed all the data that can be collected are of large interest for machine learning techniques. We now mention this just after Figure 9: “This in-depth analysis of the time course of movement and force development along with game interaction outcome allows having a clear picture of how the exergame has been played. All these data, collected over the cohort of patients served by the same hospital, can be processed through adequate machine learning techniques to: cluster patients, identify suitable models of rehabilitatoion progression for each cluster and optimize the scheduling of hospital recall visits and the therapy program [6].

Q. Moreover, the score achieved on a game is dependent not only on the use of hand/fingers but also on the gamers skill that is supposed to improve by continue playing. How the system can compensate this to estimate the improvement in hand/fingers use from the game data.

This is indeed an interesting issue. I wish to than you observation and we comment this inside the paper (paragraph at Line 484): “Indeed the score achieved on a game is dependent not only on the use of hand/fingers but also on the gamers skill that theoretically can improve by continue playing. However, this possibility is very remote here as the ability in the game is strictly dependent on the force or motion skills acquired. Therefore, this information is used to set proper level of difficulty of the exer-game, by increasing range, force or accuracy required to animate the avatar.”

Q. Another major issue is the lack of comparison between the proposed technique and reference techniques for hand/fingers rehabilitation using a trainer on-site. Have the authors made experiments, for example using the proposed system on a group of young children and standard techniques for another group to compare the performances (hand/finger rehabilitation over time etc.) between the two methods?

Classical procedures are based on building blocks used in the clinics in one-to-one sessions with the therapist, while we aim to use the toys at home under supervision of the child’s parents and, as far, as we know there are no similar systems around. Usability results were reported in [61]. Nevertheless we are in the process of filing to preparing documentation to the ethical committee of the Hospital S. Giuseppe, to start an efficacy trial whose results will be published in a clinical journal. Quantitative comparison with other systems that measure force, but are able to accommodate only one single prehension modality, is reported just after Figure 7: “The load-cell based sensing device has first been tested for accuracy, repeatability and linearity (Appendix B) to assess measurement validity following a procedure similar to that proposed in [45]. The resolution and accuracy are on average an order of magnitude larger than that of dynamometers typically used in the clinics [40–42], that have an accuracy of 50 g, with a sensitivity of 10g [40]. This is due to the fact that such dynamometers have been developed to assess grip force mainly in adults and elders and have a range that is almost twenty times larger as it reaches 90 Kg. The devices itself is also of large dimension. The focus on paediatric hand has allowed miniaturizing the device on one side and on increasing the accuracy and resolution on the other, to values that are larger than the target set by clinicians.”

Reviewer 2 Report

This paper documents a study on hand rehabilitation, as such it is an interesting research topic. However, when reading the paper I fond that there are multiple issues with the manuscript. My comments are as follows:

The general standard of English grammar is very poor with errors in both the grammar and the semantics. The use of the definite and indefinite article ai an issue along with instances of errors in word selection. The paper must be revised with respect to the use of English with careful proof reading by a native English speaker. The paper has a descriptive title, abstract, and keywords. The Introduction must be completely revised. As it currently stands it is both an introduction with a background and a very brief review of related research. The current Introduction requires separating into two dedicated sections: An Introduction with the background, the contribution clearly stated, and a paper structure outline. A Section addressing related research with a comprehensive analysis.  The references are not adequate as there are many publications on the topic. The authors need to consider alternative approaches and present a comparative analysis comparing the available approaches and the proposed method. I found the paper to be very hard to read and understand as the paper structure is not logical and there is no clear narrative to follow when reading the paper.  The paper must be revised with a logical structure and a clear narrative. The materials and methods, related research, and experiments (in the appendix) are confused as set out in the manuscript. A complete revision and restructuring of the paper is required to produce a clear and concise argument with the experiments incorporated in the body of the manuscript. I found that the materials and methods are not adequately described and discussed. This area of the study needs complete revision and extension.

In summary, the paper is potentially very interesting to the journal and the intended audience. However, as currently constituted the paper presents a confused discussion which must be revised and extended as noted in my review.

Author Response

Resubmission of the paper: "Hand rehabilitation and tele monitoring through smart toys", by N.Alberto Borghese, Jacopo Essenziale, Renato Mainetti, Elena Mancon, Rossella Pagliaro, Giorgio Pajardi, for the Special Issue on “Sensor Technologies for Caring People with Disabilities for Sensors journal.

Reply to comments of Reviewer N. 2

We report here a point-to-point reply to reviewer N.2 concerns:

 Q. The general standard of English grammar is very poor with errors in both the grammar and the semantics. The use of the definite and indefinite article ai an issue along with instances of errors in word selection. The paper must be revised with respect to the use of English with careful proof reading by a native English speaker.
A. A colleague who has been living in the United States for two years has revised the paper. In this process, all small errors should have been corrected. If this were not sufficient and the paper were accepted, we will have the paper revised by a professional English reviewer.

Q. The paper has a descriptive title, abstract, and keywords. The Introduction must be completely revised. As it currently stands it is both an introduction with a background and a very brief review of related research. The current Introduction requires separating into two dedicated sections: An Introduction with the background, the contribution clearly stated, and a paper structure outline. A Section addressing related research with a comprehensive analysis. The references are not adequate as there are many publications on the topic. The authors need to consider alternative approaches and present a comparative analysis comparing the available approaches and the proposed method. I found the paper to be very hard to read and understand as the paper structure is not logical and there is no clear narrative to follow when reading the paper. The paper must be revised with a logical structure and a clear narrative. The materials and methods, related research, and experiments (in the appendix) are confused as set out in the manuscript. A complete revision and restructuring of the paper is required to produce a clear and concise argument with the experiments incorporated in the body of the manuscript. I found that the materials and methods are not adequately described and discussed. This area of the study needs complete revision and extension.
A. We have realized that in the first draft of the paper some information has been condensed too much. We have therefore revised deeply the organization of the paper, as suggested by Reviewer N.2, as follows:

Introduction has been split into introduction and background. Introduction contains the motivation and the problem addressed (rehabilitation and telemonitoring of pediatric hand). The last paragraph summarizes the content of the different Sections. We revise the literature on the topic subdividing it into 4 categories: robotics, camera trackers, tablet and smart objects. The final paragraph clearly marks what is still missing in these approaches. Materials and Methods is now organized according to a clear path: functional specifications definition, implementation of the platform and on a flexible sensing device in the for of a toy, the general architecture and the exergames defined. Section on specifications has been streamlined, with a clear definition of the methodology and of the exercises identified. A comparison of the platform with existing methods both in terms of accuracy and usability. Conclusion that stresses the role of smart objects for telemonitoring and telerehabilitation. Appendix has been split into Appendix A (Amplification design) and Appendix B (Characterization of force measurements). These contains details that the reader might be interested in. However, now the results of the Appendixes are fully reported and discussed in the main text.

As reorganization and rewriting has been very heavy we could not use track change Word functionality.

Moreover, the text was proof-read by a colleague of us who has been living in the United States for two years. In this process, all small errors should have been corrected. If this were not sufficient and the paper were accepted, we will have the paper revised by a professional English reviewer.

Round 2

Reviewer 1 Report

The authors revised the manuscript according to the reviewers' comments. I think the paper can be published after minor revisions.

1) At lines 257-265 it is described the crocodile shaped button used as a controller in the exergames. The controller sends binary information (on/off) to the smartphone/tablet via Bluetooth and it is powered by battery. What type of battery is used? What is the expected duration of the battery? Can the battery be recharged (for example by USB) or it must be replaced?

2) Lines 333, 340 and 346: "Figure a", "Figure b" and "Figure c" should be "Figure 6a", "Figure 6b" and "Figure 6c".

3) Line 518: "Figure 1d". I think there is an error in the reference to the figure. Maybe it is "Figure 2d".

Author Response

Resubmission of the paper: "Hand rehabilitation and tele monitoring through smart toys", by N. Alberto Borghese, Jacopo Essenziale, Renato Mainetti, Elena Mancon, Rossella Pagliaro, Giorgio Pajardi, for the Special Issue on “Sensor Technologies for Caring People with Disabilities for Sensors journal.

Reply to comments of Reviewer N. 1

Q. The authors revised the manuscript according to the reviewers' comments. I think the paper can be published after minor revisions.

A. Thank you. Indeed suggestions were very good but challenging and required a lot of work.

Q. At lines 257-265 it is described the crocodile shaped button used as a controller in the exergames. The controller sends binary information (on/off) to the smartphone/tablet via Bluetooth and it is powered by battery. What type of battery is used? What is the expected duration of the battery? Can the battery be recharged (for example by USB) or it must be replaced?

A. We have added the required information on the battery. This is surely an issue for all smart objects and we thank the reviewer for having raised this issue. We use a source of energy of different capacity is used for the two pressure sensing device. This information is added and discussed in the Discussion Section (lines 410-417): “A single non-rechargeable button cell (CR2032 3V) is used in the crocodile sensor. This is dimensioned to operate the device for one month half an hour a day before the cell discharges and it has to be replaced. The opening in the lower jaw (Figure 7d) makes this operation easy. The Lego cottage accommodates a larger source of energy inside the case. We use here a pair of rechargeable stylus batteries (1,5 V AA) that are recharged typically at the end of the weak. Indeed rechargeable modalities that do not require extracingt the batteries from their housing (USB, induction) would be preferable but they require additional room for the components and will be part of a next refinement stage.”

Q. Lines 333, 340 and 346: "Figure a", "Figure b" and "Figure c" should be "Figure 6a", "Figure 6b" and "Figure 6c".

A. Thank you. Sometimes, Word cross-references induce in error.

Q. Line 518: "Figure 1d". I think there is an error in the reference to the figure. Maybe it is "Figure 2d".

A. Indeed! Thank for spotting this error. As you imagine, the figures order was changed in the main text and this reference of the Appendix was not updated.

Reviewer 2 Report

The authors have made considerable effort in. presenting what is in effect a new paper. The English changes (almost the complete paper) have been noted and generally the standard of English grammar is acceptable. I noted that the authors will submit the paper for professional English checking if accepted, it is quite hard to discern the English grammar given the extensive revisions (revision changes shown) so when the final manuscript is complete this may be a sensible course of action to address any (albeit minor) English corrections needed. 

My comments:

I noted the changes to the Introduction to show an Introduction and Background (possible better entitled Related Research? There is a formatting error in pages 8/9 (large white space) which may be addressed at the proofing stage. An improvement which is needed is the identify open research questions (ORQ)(inevitable in such a study). The ORQ will be both medical and computational. Such ORQ will form the basis for future directions for research which should be introduced into the paper. This will provide interesting observations by the authors based on their knowledge of the topic resulting resulting from their research study.

In general, I am content with the revisions which have in my view addressed many of my previous concerns. However, in completing the revised manuscript, the authors have raised some questions (see my comments) which require addressing.

I think the article will be of interest to the intended audience and the journal, in addressing my comments the authors will increase the potential readership (the audience) for their paper and also improve the presentation of the study. 

Author Response

Resubmission of the paper: "Hand rehabilitation and tele monitoring through smart toys", by N.Alberto Borghese, Jacopo Essenziale, Renato Mainetti, Elena Mancon, Rossella Pagliaro, Giorgio Pajardi, for the Special Issue on “Sensor Technologies for Caring People with Disabilities for Sensors journal.

Reply to comments of Reviewer N. 2

Q. The authors have made considerable effort in. presenting what is in effect a new paper. The English changes (almost the complete paper) have been noted and generally the standard of English grammar is acceptable. I noted that the authors will submit the paper for professional English checking if accepted, it is quite hard to discern the English grammar given the extensive revisions (revision changes shown) so when the final manuscript is complete this may be a sensible course of action to address any (albeit minor) English corrections needed. 

A. Thank you. Indeed reviewers suggestions were very good but challenging and have required a lot of work. We are glad that this was appreciated.

Q. I noted the changes to the Introduction to show an Introduction and Background (possible better entitled. Related Research?

A. We have taken your suggestion and modified the heading.

Q. There is a formatting error in pages 8/9 (large white space) which may be addressed at the proofing stage. An improvement which is needed is the identify open research questions (ORQ)(inevitable in such a study). The ORQ will be both medical and computational. Such ORQ will form the basis for future directions for research which should be introduced into the paper. This will provide interesting observations by the authors based on their knowledge of the topic resulting from their research study.

Q. In general, I am content with the revisions which have in my view addressed many of my previous concerns. However, in completing the revised manuscript, the authors have raised some questions (see my comments) which require addressing.
A. We have added ORQ at the end of the Discussion Session (lines 505-521). We hope to have interpreted well the request by the reviewer.

“The platform can acquire a huge amount of data from a large population of patients. This enables machine learning algorithms to browse the data to extract meaningful information at the population level. One of the most promising direction is the identification of homogenous clusters of patients for whom a tailored rehabilitation program can be defined and the frequency of recall visits can be optimized. The same data can allow the comparison of different rehabilitation programs to identify strong and weak elements.

The large population could be also the basis for creating a virtual community of patients and their parents. This can, on one side, support gamification to boost children motivation in performing the exercises (e.g. virtual medals, points, leader boards, ranking and so forth, cf. [71]), on the other side it connects all the stakeholders and provides them a place where to exchange information, data, suggestions and where they can find continuous support from peers.

Some additional work can be spent in trying to identifying automatically when the child has made the right movement. In fact, children tend to use either their intact fingers or their intact hand to perform the movements required by the exergames; presently their parents correct them to perform the right movement. This could be achieved, for instance, by analysing the pressure or movement time course with adequate classification algorithms. Finally, although the hardware is relatively simple and cheap (at present less than 100 Euros), its cost should be further reduced to enable massive diffusion.”

Q. I think the article will be of interest to the intended audience and the journal, in addressing my comments the authors will increase the potential readership (the audience) for their paper and also improve the presentation of the study. 
A. Thank you for your comments and criticisms that were indeed pertinent and relevant.